# Palmitoylation of KSHV pORF55 is required for Golgi localization and efficient progeny virion production

Yaru Zhou[1,2,3], Xuezhang Tian[1,2], Shaowei Wang[1,2], Ming Gao[1,2], Chuchu Zhang[1,2], Jiali Ma[1,2], Xi Cheng[1,2], Lei Bai[4], Hai-Bin Qin[5,6], Min-Hua Luo[5,6], Qingsong Qin[7], Baishan Jiang[2], Ke Lan[2,4], Junjie Zhang[1,2,3]*

1 State Key Laboratory of Oral & Maxillofacial Reconstruction and Regeneration, Key Laboratory of Oral Biomedicine Ministry of Education, Hubei Key Laboratory of Stomatology, School & Hospital of Stomatology, State Key Laboratory of Virology, Medical Research Institute, Wuhan University, Wuhan, China, 2 Frontier Science Center for Immunology and Metabolism, Medical Research Institute, Wuhan University, Wuhan, China, 3 Hubei Key Laboratory of Tumor Biological Behaviors, Hubei Province Cancer Clinical Study Center, Zhongnan Hospital of Wuhan University, Wuhan, China, 4 State Key Laboratory of Virology, School of Life Sciences, Wuhan University, Wuhan, China, 5 State Key Laboratory of Virology, CAS Center for Excellence in Brain Science and Intelligence Technology, Center for Biosafety Mega-Science, Wuhan Institute of Virology, Chinese Academy of Sciences, Wuhan, China, 6 University of Chinese Academy of Sciences, Beijing, China, 7 Laboratory of Human Virology and Oncology, Shantou University Medical College, Shantou, China

* junjiezhang@whu.edu.cn

**Data Availability Statement:** All the relevant date are in the manuscript and its supporting information files.

## Abstract

Kaposi's sarcoma-associated herpesvirus (KSHV) is a double-stranded DNA virus etiologically associated with multiple malignancies. Both latency and sporadic lytic reactivation contribute to KSHV-associated malignancies, however, the specific roles of many KSHV lytic gene products in KSHV replication remain elusive. In this study, we report that ablation of ORF55, a late gene encoding a tegument protein, does not impact KSHV lytic reactivation but significantly reduces the production of progeny virions. We found that cysteine 10 and 11 (C10 and C11) of pORF55 are palmitoylated, and the palmytoilation is essential for its Golgi localization and secondary envelope formation. Palmitoylation-defective pORF55 mutants are unstable and undergo proteasomal degradation. Notably, introduction of a putative Golgi localization sequence to these palmitoylation-defective pORF55 mutants restores Golgi localization and fully reinstates KSHV progeny virion production. Together, our study provides new insight into the critical role of pORF55 palmitoylation in KSHV progeny virion production and offers potential therapeutic targets for the treatment of related malignancies.

## Author summary

Kaposi's sarcoma-associated herpesvirus (KSHV) is an oncogenic herpesvirus associated with multiple human malignancies. Nonetheless, the roles of numerous viral proteins in the viral life cycle remain inadequately characterized. Employing CRISPR knockout screening, we identified the viral tegument protein pORF55 as pivotal in the production

**Funding:** J.Z. is supported by the National Key Research and Development Program of China (2023YFC2306600), grants from National Natural Science Foundation of China (82372241, 82172261, and 31970156), and the Fundamental Research Funds for the Central Universities (2042022dx0003, 2042023kf0235). Q.Q. is supported by the Open Research Fund Program of the State Key Laboratory of Virology of China (2022KF010). The funders had no role in study design, data collection and analysis, decision to publish, or preparation of the manuscript.

**Competing interests:** The authors have declared that no competing interests exist.

of infectious progeny virions. We found that pORF55 is palmitoylated at cysteine 10 and 11, which is required for its Golgi localization. Palmitoylation-deficient mutants of pORF55 are unstable and fail to support secondary envelopment formation, a critical step for viral assembly and egress. Interestingly, we found that forced restoration of the Golgi localization of the palmitoylation-deficient pORF55 mutants completely reinstates the infectious progeny virion production. Hence, our study underscores the central role of Golgi localization resulting from pORF55 palmitoylation. Our study not only elucidates the role of pORF55 in viral replication, but also suggests targeting its palmitoylation as a potential therapeutic strategy for curtailing viral replication and treating related pathogenesis.

## Introduction

Kaposi's sarcoma-associated herpesvirus (KSHV), also referred to as human herpesvirus 8, belongs to the γ herpesvirus family, and is etiologically associated with multiple human malignances, including Kaposi's sarcoma (KS), KSHV-associated inflammatory cytokine syndrome (KICS), and two lymphoproliferative diseases, primary effusion lymphoma (PEL) and multicentric Castleman's disease (MCD) [1–5]. The life cycle of KSHV contains two distinct phases: latency and lytic phase [6,7]. KSHV establishes latency in primary infected cells to evade host immune system, and initiates lytic reactivation under certain stimulations such as hypoxia, oxidative stress and sodium butyrate treatment [8]. Previous studies have established that, in addition to latency, the lytic phase also contributes to KSHV-associated malignancies [9,10]. Substantial efforts have been made to uncover how viral genes regulate lytic replication and their roles in viral pathogenesis [6,8]. These studies have established that the replication and transcription activator (RTA) encoded by KSHV ORF50 functions as a master transcription factor controlling the latent-lytic switch, and is also responsible for the activation of numerous early and late genes [8]. These lytic genes have diverse functions in various aspects of the viral life cycle, including viral gene transcription, RNA splicing, protein expression and assembly [8,11,12]. However, KSHV is a large DNA virus encoding more than 80 viral proteins, many of which remain poorly characterized.

pORF55 is a tegument protein of KSHV, and has been shown to interact with KSHV vBCL2 and facilitate KSHV virion assembly [13]. However, the specific role of pORF55 in KSHV lytic phase remains poorly understood. ORF55 is conserved among all herpesviruses, and its homologue in α-herpesvirus HSV-1, known as UL51, has been best studied [14]. These studies reveal that UL51 forms a complex with UL7 (a homologue of KSHV pORF42) at the Golgi apparatus [15]. Notably, palmitoylation of the N-terminal Cysteine 9 (C9) of UL51 is required for its Golgi localization [14, 16]. Structural studies on the UL51-UL7 complex of HSV-1 and the BSRF1-BBRF2 (homologues of KSHV pORF55-pORF42) complex of EBV have proposed a model whereby these complexes bridge viral glycoproteins and the viral capsid during the late lytic phase, thereby facilitating secondary envelopment and assembly of virion particles [15,17]. While the pORF55-pORF42 complex of KSHV similarly localizes at the Golgi [15], the specific role of pORF55 in KSHV replication remains elusive.

In this study, we report that KSHV pORF55 undergoes palmitoylation at C10 and C11, and the palmytoilation is required for its Golgi localization and secondary envelope formation. Introduction of C10S or C11S mutations into ORF55 of the KSHV genome greatly impairs progeny virion production. These palmitoylation-defective pORF55 mutants fail to localize to the Golgi and undergo proteasomal degradation. Importantly, the introduction of a putative

Golgi targeting sequence to these palmitoylation-defective pORF55 mutants fully restores their Golgi localization and progeny virion production. Together, our study provides new insights into the critical role of pORF55 palmitoylation in KSHV progeny virion production and offers potential therapeutic targets for the treatment of related malignancies.

## Results

### CRISPR knockout screen identifies the critical role of ORF55 in KSHV lytic replication

To investigate the role of KSHV genes in lytic reactivation, we chose less-studied KSHV lytic genes and designed sgRNAs to target them. To validate our approach, we first designed an sgRNA targeting ORF50, which encodes the essential transcription factor RTA for KSHV lytic reactivation [7,8]. As expected, deletion of ORF50 nearly abolished KSHV virion production (Fig 1A). We then designed an sgRNA targeting ORF74, which encodes vGPCR and is involved in viral oncogenesis but not viral replication [9,18,19]. As anticipated, ablation of ORF74 had no impact on progeny virion production (Fig 1B). These results confirm the viability of our CRISPR knockout strategy for identifying viral genes involved in lytic reactivation. Our small-scale CRISPR screen revealed that numerous viral genes contribute to KSHV lytic reactivation, among which knockout of ORF55 had the most significant impact on KSHV reactivation (Fig 1C). We then designed two independent sgRNAs that could efficiently deplete ORF55 (Fig 1D), and verified that sgRNA-mediated knockout of ORF55 greatly reduced KSHV progeny virion production (Fig 1E). These results indicate that ORF55 plays a critical role in KSHV lytic phase.

### Characterization of BAC-derived ΔORF55 KSHV

To investigate the role of ORF55 in KSHV infection, we knocked out ORF55 in KSHV genome using an infectious bacterial artificial chromosome (BAC) clone [20,21]. Following lytic induction, immunoblotting analysis confirmed the successful deletion of ORF55 in SLK.iBAC cells (hereafter referred to as SLK.iBACΔORF55) (Fig 2A). Knockout of ORF55 resulted in a remarkable more than 20-fold reduction in the production of progeny virions (Fig 2B). The transcription of viral lytic genes, including ORF50, ORF57, and ORF25, was not affected by pORF55 deficiency (Fig 2C). Similarly, the protein levels of RTA, pORF45, and pORF57 were comparable between SLK.iBAC and SLK.iBACΔORF55 upon lytic induction (Fig 2D). These findings indicate that pORF55 does not influence viral gene transcription and protein expression following lytic reactivation. Subsequently, we quantified viral genome copy number and found that pORF55 deficiency had no impact on intracellular viral genome copy number. However, extracellular viral genome copy number was markedly reduced in SLK.iBACΔORF55, indicating that pORF55 does not influence viral genome replication but may play a role in viral assembly and release (Fig 2E and 2F). These results collectively indicate that pORF55 is not involved in KSHV lytic reactivation but plays a critical role in progeny virion production.

### pORF55 is localized to the Golgi apparatus

Previous studies have shown that UL51, the HSV-1 homologue of pORF55, localizes to the Golgi [16,22]. To characterize the subcellular localization of pORF55, we conducted immunofluorescence staining and observed that transiently expressed pORF55 effectively co-localized with the Golgi marker GM130, but not the ER marker Calnexin (Fig 3A and 3B). Furthermore, we induced lytic reactivation in SLK.iBAC cells and found that pORF55 was also localized at the Golgi (Fig 3C). These data indicate that pORF55 is localized to the Golgi apparatus.

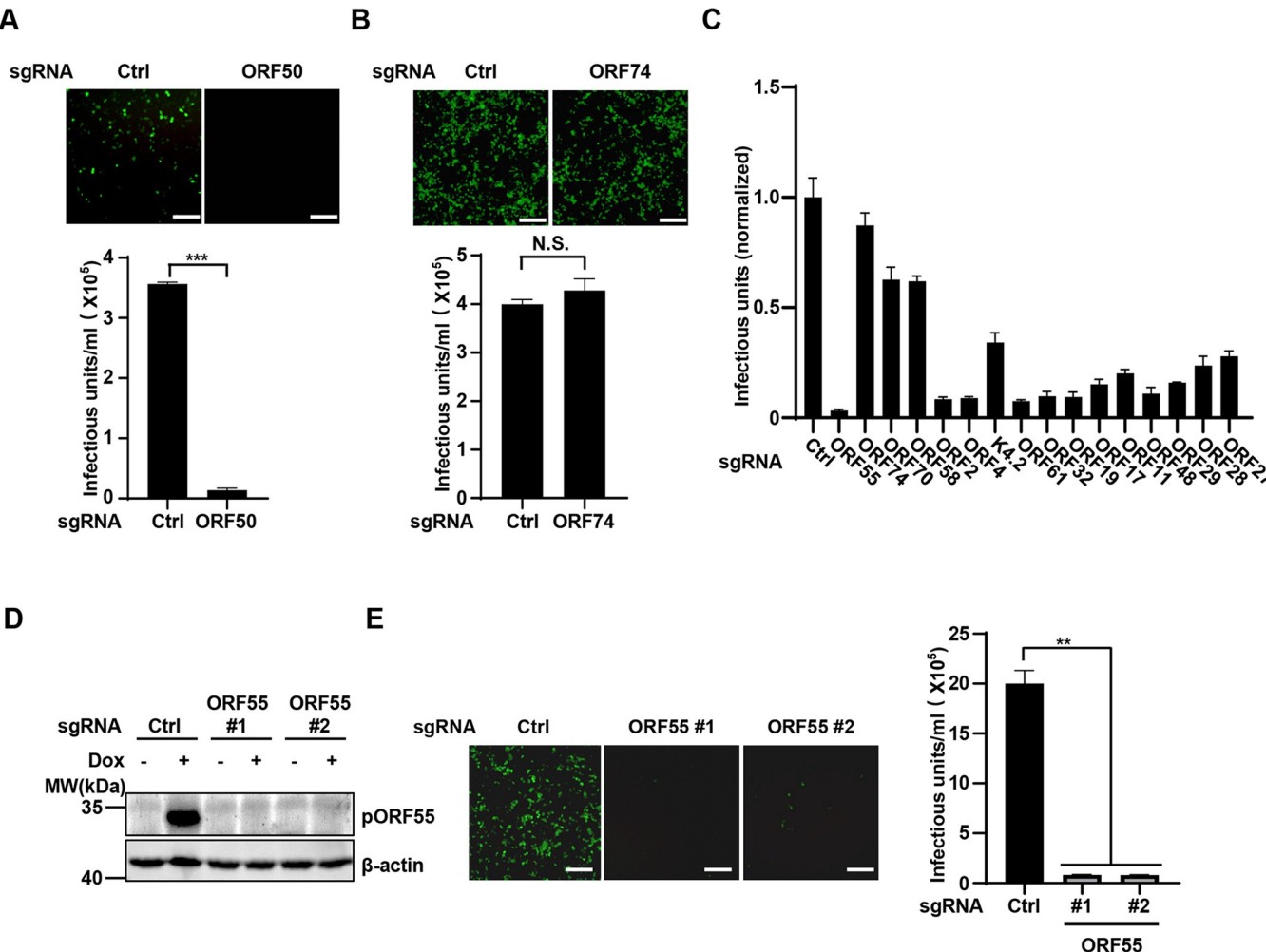

**Fig 1. CRISPR-Cas9 knockout screen reveals the critical role of ORF55 in KSHV lytic replication.** (A and B) SLK.iBAC cells were transduced with control sgRNA or sgRNA targeting ORF50 or ORF74 to generate stable cells. The cells were induced with Dox (1 μg/mL) and sodium butyrate (0.5 mM) for 48 h. The supernatants were collected to infect HEK293T cells, and GFP expression was imaged at 24 h post infection. Scale bars, 100 μm. KSHV infectious units were calculated based on flow cytometry analysis of GFP-positive cell percentage. (C) SLK.iBAC cells were transduced with sgRNA targeting the indicated viral genes to generate stable cells. KSHV infectious units were calculated as described in Fig 1A. (D-E) SLK.iBAC cells were stably transduced with control sgRNA or sgRNA targeting ORF55. The stable cells were induced with Dox (1 μg/mL) and sodium butyrate (0.5 mM) for 48 h, and whole cell lysates (WCLs) were analyzed by immunoblotting (D). KSHV infectious units were calculated as described in Fig 1A (E). Scale bars,100 μm.

## Palmitoylation of pORF55 is required for its Golgi localization

Based on previous studies about HSV-1 UL51, we speculate that pORF55 may also be palmitoylated and the palmitoylation is crucial for its Golgi localization. To investigate this, we treated pORF55-expressing cells with a widely used palmitoylation inhibitor, 2-BP. Intriguingly, while pORF55 was localized to the Golgi, 2-BP treatment completely disrupted its Golgi localization (Fig 4A). Furthermore, 2-BP treatment impaired progeny virion production in SLK.iBAC WT cells upon lytic reactivation. However, 2-BP treatment could not further reduce infectious virion production in SLK.iBACΔORF55 cells, likely due to the already diminished virion production of the ΔORF55 mutant (Fig 4B). Ganciclovir, a widely used herpesvirus replication inhibitor, was included to assess the impact on virion production, and the results indicate that both ORF55-KO and 2-BP treatment more severely inhibited KSHV virion production compared with Ganciclovir treatment (S1A Fig). Moreover, treatment with both

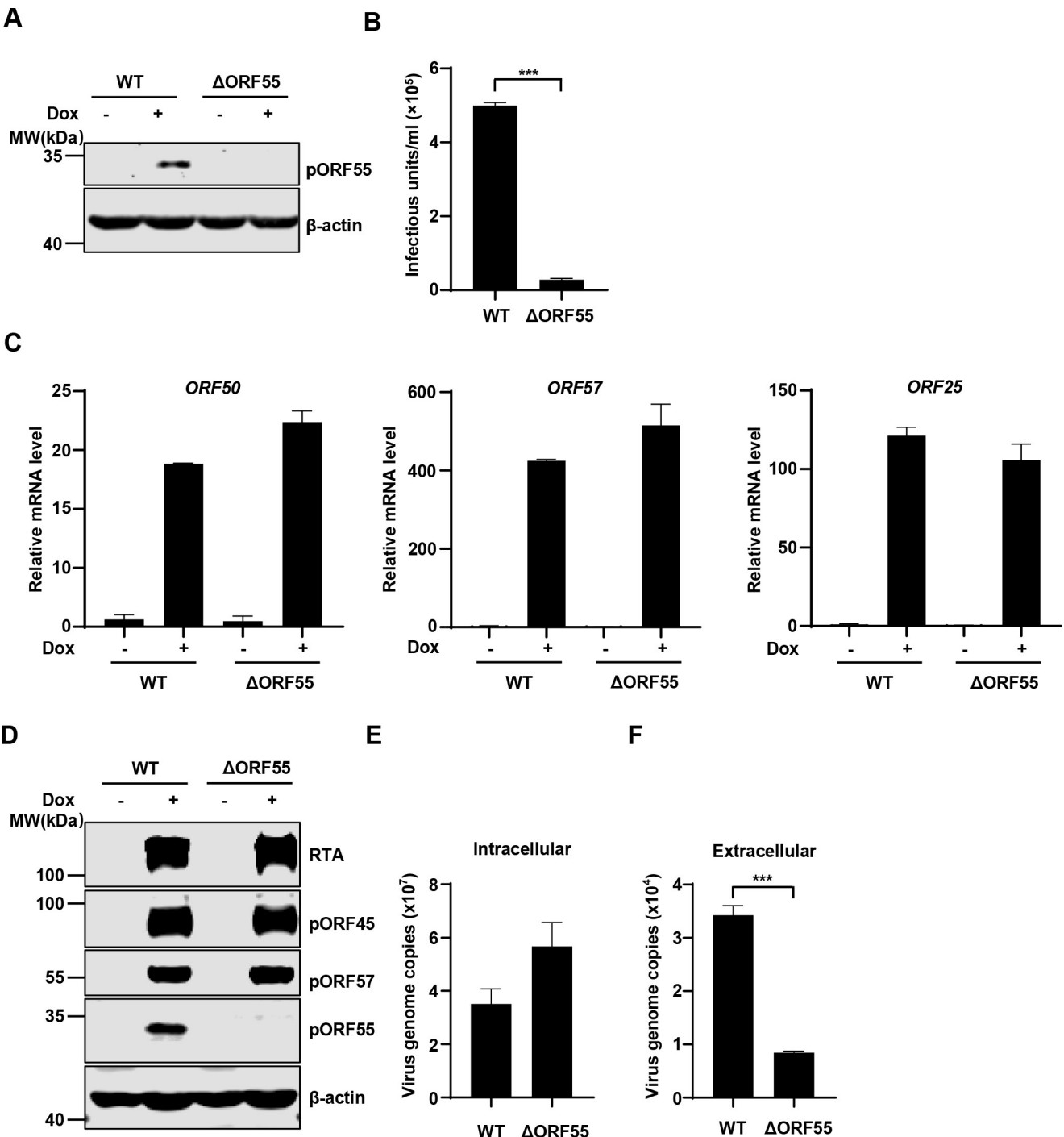

**Fig 2. KSHV ORF55 has a critical role in progeny virion production.** (A-B) SLK.iBAC or SLK.iBACΔORF55 cells were induced with Dox (1 μg/mL) and sodium butyrate (0.5 mM) for 48 h, and WCLs were analyzed by immunoblotting (A). The supernatants were collected, and KSHV infectious units were quantified (B). (C-F) SLK.iBAC or SLK.iBACΔORF55 cells were induced as described in Fig 2A. Viral gene expression was quantified by RT-qPCR (C), and WCLs were analyzed by immunoblotting (D). The intracellular (E) and extracellular (F) viral genome copy number of KSHV-WT or KSHV-ΔORF55 were quantified by qPCR.

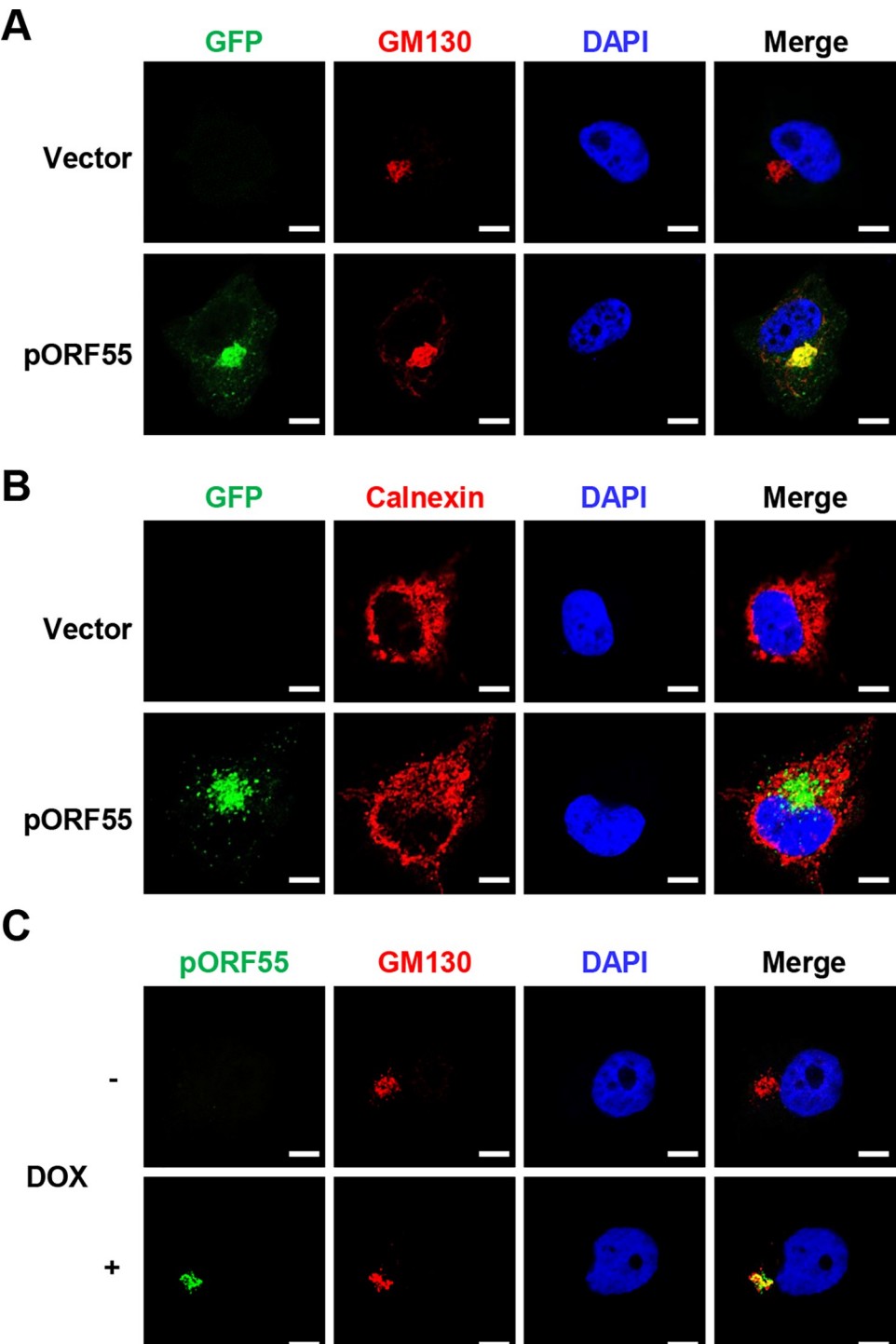

**Fig 3. pORF55 is localized to the Golgi apparatus.** (A) Hela cells were transfected with vector control or ORF55-EGFP, followed by immunofluorescence staining with GM130 as a Golgi marker. (B) Hela cells were transfected with vector control or ORF55-EGFP, followed by immunofluorescence staining with Calnexin as an ER marker. (C) SLK.iBAC cells were induced with Dox (1 μg/mL) for 36 h, followed by Immunofluorescence staining with antibodies against pORF55 and GM130. The nuclei were counterstained with DAPI in all panels. Scale bars,10 μm.

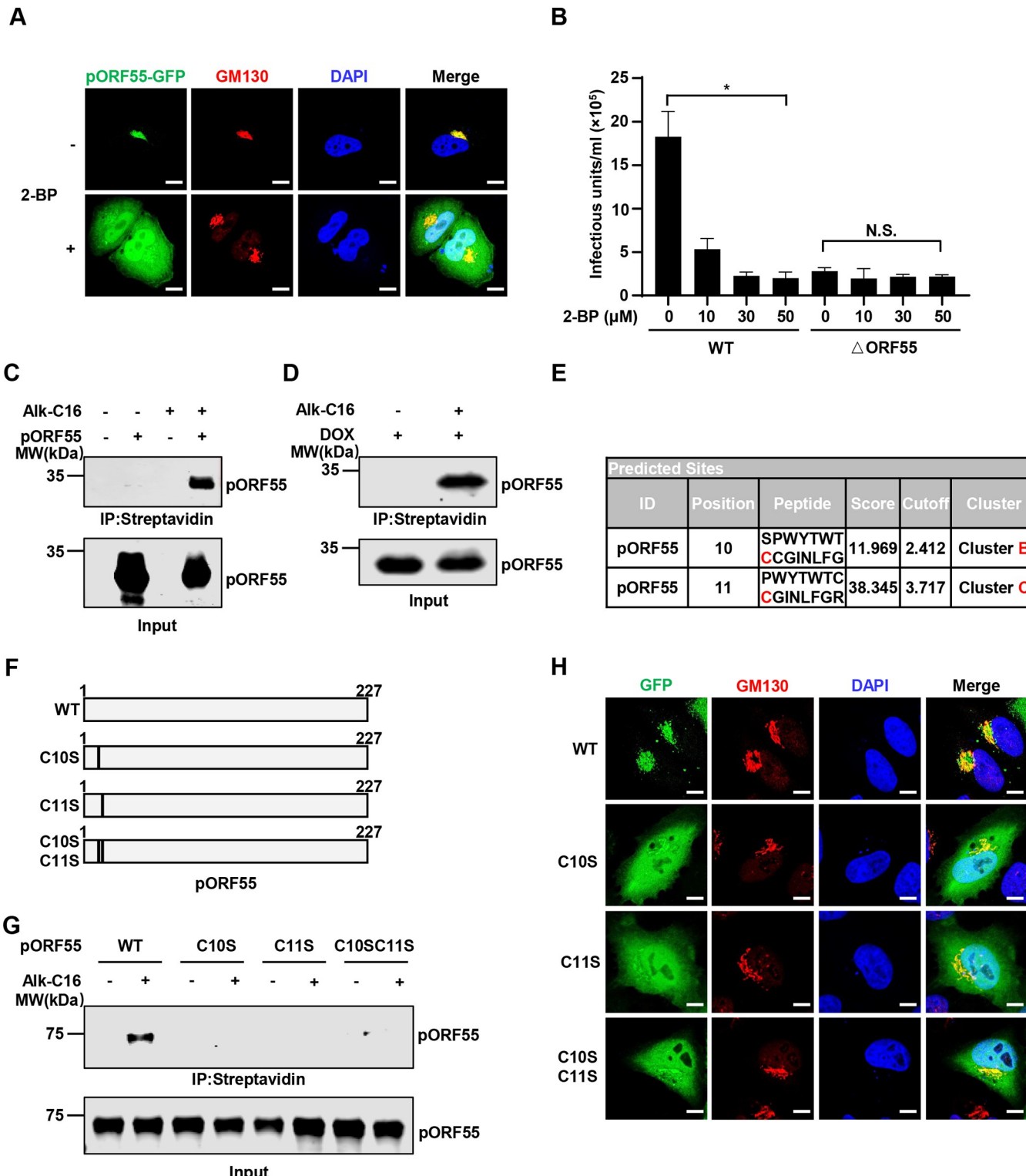

**Fig 4. Palmitoylation of pORF55 is necessary for its Golgi localization.** (A) Hela cells expressing pORF55-EGFP were treated with control solvent or 2-BP (30 μM) for 16 h, followed by immunofluorescence staining. The nuclei were counterstained with DAPI. Scale bars,10 μm. (B) SLK.iBAC or SLK.iBACΔORF55 were induced with Dox (1 μg/mL) and sodium butyrate (0.5 mM) in the presence of the indicated amounts of 2-BP. KSHV infectious units were quantified at 48 h post-induction. (C) HEK293T cells were transfected with FLAG-ORF55 for 24 h, and then labeled with Alk-C16 (100 μM) for 8 h. Cell lysates were collected and subjected to click chemistry by reacting with an azide-functionalized biotin, followed by immunoprecipitation with streptavidin beads. The input and immunoprecipitated proteins were detected with an anti-pORF55 polyclonal antibody. (D) SLK.iBAC cells were induced with Dox (1 μg/mL) for 24 h, and

click chemistry was used to detect pORF55 palmitoylation as described in Fig 4C. (E) The palmitoylation modification sites of pORF55 were predicted using CSS-Palm 4.0. (F) Schematic diagram of the pORF55 mutants. (G) Click chemistry was employed to detect pORF55 palmitoylation in HEK293T cells transfected with FLAG-ORF55 and the mutants. (H) Hela cells were transfected with ORF55-EGFP or the mutants, and immunofluorescence staining were performed with an antibody against GM130 (a Golgi marker). The nuclei were counterstained by DAPI. Scale bars,10 μm.

Ganciclovir and 2-BP showed minimal cell toxicity (S1B Fig). These data suggest that palmitoylation of pORF55 is required for its Golgi localization and efficient virion generation. In line with previous studies on UL51 [16], we found that the N-terminal 35 amino acids (aa) of pORF55, when fused to EGFP, was sufficient for Golgi targeting. By contrast, deletion of the N-terminal 35aa or even 15aa abolished the Golgi localization of pORF55 (S1C and S1D Fig). These results indicate that the palmitoylation sites of pORF55 are likely localized in the N-terminal 15aa.

Next, we employed a well-established click chemistry method to determine whether pORF55 is palmitoylated with Alk-C16 labeling [23–25]. Both ectopically expressed pORF55 and pORF55 induced during viral lytic reactivation exhibited robust Alk-C16 labeling, indicating a strong palmitoylation of pORF55 (Fig 4C and 4D). We then sought to pinpoint the palmitoylation sites of pORF55. To achieve that, we employed GPS-Palm to predict the palmitoylation sites of pORF55, and two prominent sites (cysteine 10 and 11) stood out as potential palmitoylation sites (Fig 4E) [26–28]. Notably, Alk-C16 labeling was completely abolished when we introduced C10S, C11S, or C10S/C11S into pORF55, suggesting that C10 and C11 of pORF55 are palmitoylated, and the stable palmitoylation of C10 and C11 is interdependent (Fig 4F and 4G). Consistently, immunofluorescence staining revealed that C10S, C11S and C10S/C11S mutants of pORF55 were unable to localize specifically to the Golgi (Figs 4H and S1E). Together, these results indicate that palmitoylation of pORF55 at C10 and C11 is essential for its Golgi localization.

## Palmitoylation of pORF55 is essential for secondary envelopment and efficient virion production

To explore the role of pORF55 palmitoylation in KSHV infection, we reconstituted SLK.iBAC∆ORF55 with WT pORF55 or the palmitoylation-defective mutants, including C10S, C11S and C10SC11S. The re-expression of WT pORF55, but not the palmitoylation-defective mutants (C10S, C11S and C10SC11S), could rescue the impaired virion production in SLK.iBAC∆ORF55 (S2A Fig). Consistent with the notion that pORF55 palmitoylation is not required for lytic reactivation, we observed that viral gene transcription (ORF50, ORF57, and ORF25), viral protein expression (RTA, pORF45, and pORF57), as well as intracellular viral genome copy number were comparable between SLK.iBAC∆ORF55 reconstituted with WT pORF55 or the palmitoylation-defective mutants (S2B–S2D Fig). By contrast, the reduced extracellular viral genome copy number of SLK.iBAC∆ORF55 could be fully restored by the expression of WT pORF55, but not the palmitoylation-defective mutants (S2E Fig). These results suggest that pORF55 palmitoylation at C10 and C11 plays a critical role in efficient virion production.

To further validate the conclusion, we introduced point mutations into KSHV genome by BAC-mediated mutagenesis and generated SLK.iBAC.ORF55-C10S, ORF55-C11S, and ORF55-C10SC11S. Notably, the palmitoylation-defective mutations of pORF55 phenocopied ORF55 knockout upon the induction of lytic reactivation, as indicated by reduced infectious virion production (Fig 5A), unaffected viral gene transcription (Fig 5B), and viral protein expression (Fig 5C). Moreover, the introduction of C10S, C11S or C10SC11S into KSHV genome did not influence intracellular viral genome replication, but significantly diminished

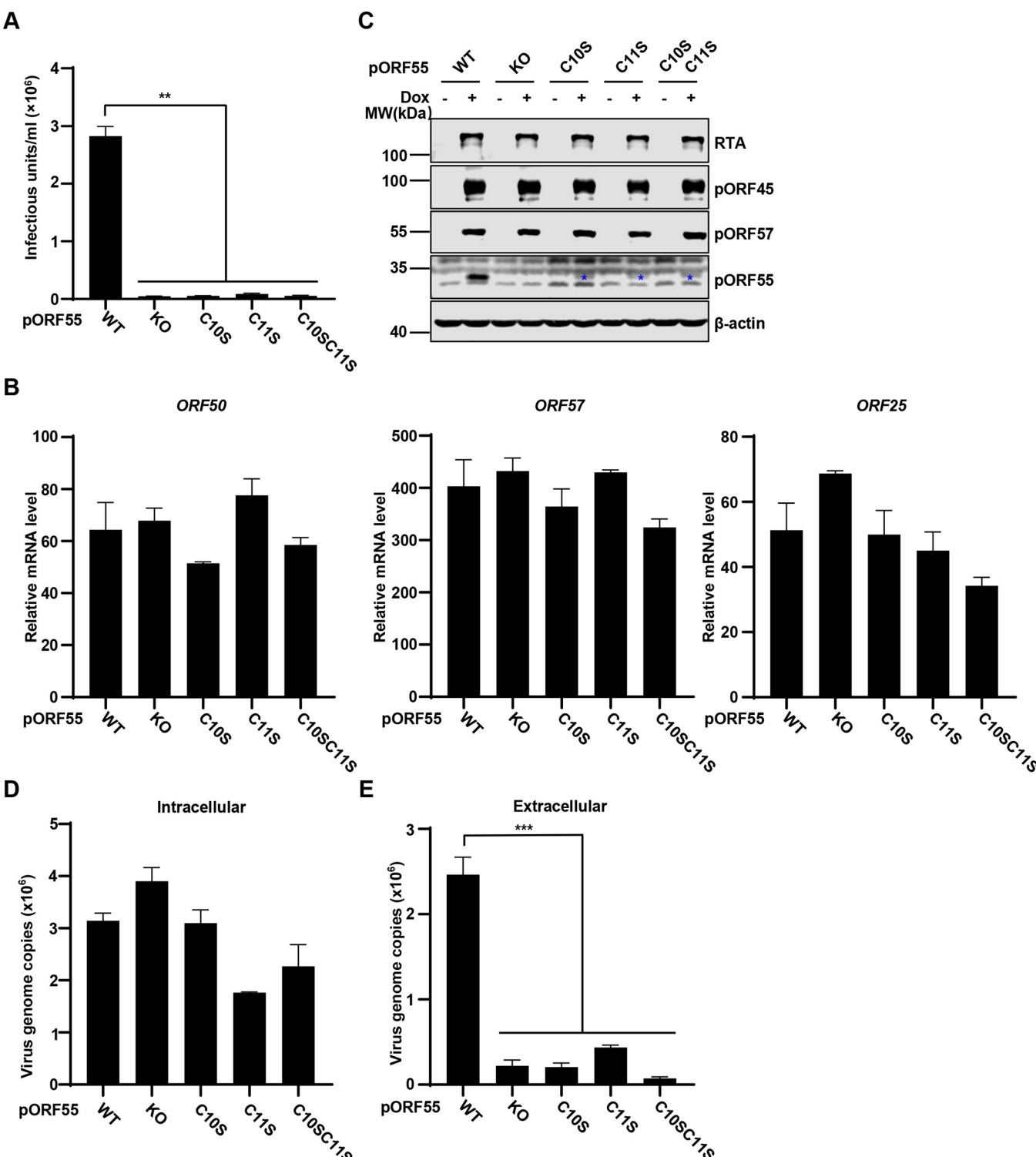

**Fig 5. Palmitoylation of pORF55 is crucial for efficient virion production.** (A-D) SLK.iBAC cells containing WT ORF55, ΔORF55 or the indicated mutants were induced with Dox (1 μg/mL) and sodium butyrate (0.5 mM) for 48 h, and KSHV infectious units in the supernatants were quantified (A). Viral gene transcription was quantified by RT-qPCR (B), and WCLs were analyzed by immunoblotting (C). The intracellular viral genome copy number was determined by qPCR analysis (D). (E) The cell lines as described in Fig 5A was induced with Dox (1 μg/mL) and sodium butyrate (0.5 mM) for 72 h, and the extracellular viral genome copy number was determined by qPCR.

extracellular viral genome copy number (Fig 5D and 5E). In addition, transmission electron microscope (TEM) analysis showed that viral capsids were packaged into specialized vesicles in WT cells, but these vesicles were not observed in ΔORF55 and the palmitoylation-deficient mutant cells. These results provide morphological evidence suggesting that pORF55 are critical for secondary envelope formation, and ORF55 KO or the palmitoylation-deficient mutants were unable to support secondary envelopment (S3 Fig). Together, these data indicate that palmitoylation of pORF55 is essential for secondary envelopment and efficient virion production.

## The Golgi localization of pORF55, mediated by its palmitoylation, is crucial for efficient virion production

We next asked how defective palmitoylation of pORF55 affects its localization. We employed immunofluorescence to monitor the localization of WT pORF55 and the palmitoylation-defective mutants. The immunofluorescence results clearly showed that the C10S, C11S, and C10SC11S mutants of pORF55 failed to localize to the Golgi, indicating that palmitoylation of pORF55 is necessary for its Golgi localization (Fig 6A and 6B). Next, we sought to redirect the palmitoylation-defective mutants of pORF55 to the Golgi and examine whether this could restore their function in progeny virion production. To achieve that, we turned to a well-established Golgi targeting sequence, altORF [29], and confirmed that GFP fused with altORF completely co-localized with the Golgi, similar to pORF55 (S4A Fig). Furthermore, pORF55Δ-N15aa, distributed throughout the entire cell, could be retargeted to the Golgi, when fused with altORF (S4B Fig). Indeed, fusion with altORF successfully redirected the palmitoylation-defective mutants of pORF55 to the Golgi (Figs 6A, 6B and S4C). Importantly, the re-expression of altORF-pORF55-C10S, -C11S, and -C10SC11S, but not the palmitoylation-defective mutants, fully restored the production of infectious virions that was significantly impaired by pORF55 deficiency (Fig 6C). In line with our earlier results, the reconstitution of altORF-fused pORF55 mutants did not impact the expression of viral proteins (RTA, pORF45 and pORF57) (Fig 6D). These data collectively indicate that the Golgi localization of pORF55, mediated by its palmitoylation, is crucial for efficient progeny virion production.

## Palmitoylation-defective pORF55 mutants demonstrate reduced stability

Throughout the study, we consistently observed that the protein levels of palmitoylation-defective pORF55 mutants (C10S, C11S and C10SC11S) were reduced compared with that of WT pORF55 (Figs 5C and S2C). We speculate that these pORF55 mutants have reduced stability since they are unable to localize to the Golgi. Treatment with the proteasome inhibitor MG132, but not the lysosome inhibitor Bafilomycin A1 (Baf-A1), could restore the expression of the pORF55 mutants (Fig 7A), suggesting that the palmitoylation-defective pORF55 mutants undergo proteasomal degradation. Consistently, the palmitoylation-defective pORF55 mutants displayed reduced half-life compared with WT pORF55 (Fig 7B). Next, we determined ubiquitin (Ub) and K48 Ub modification status of WT pORF55 and the mutants. We found that these modifications were comparable between WT pORF55 and the palmitoylation-defective mutants (Fig 7C and 7D). These results suggest that pORF55 undergoes extensive ubiquitination regardless of its Golgi localization. However, palmitoylated and thus Golgi-localized pORF55 is resistant to proteasomal degradation. Next, we tested two widely used proteasome inhibitors, MG132 and Bortezomib in KSHV lytic replication. Virion production was potently inhibited in SLK.iBAC WT, SLK.iBACΔORF55, as well as SLK.iBAC cells expressing palmitoylation-deficient mutants of ORF55 (C10S, C11S, and C10SC11S) upon treatment with both inhibitors (S5A Fig). Treatment with both MG132 and Bortezomib also suppresses viral protein expression (S5B Fig), suggesting that the proteasome system has a critical role in

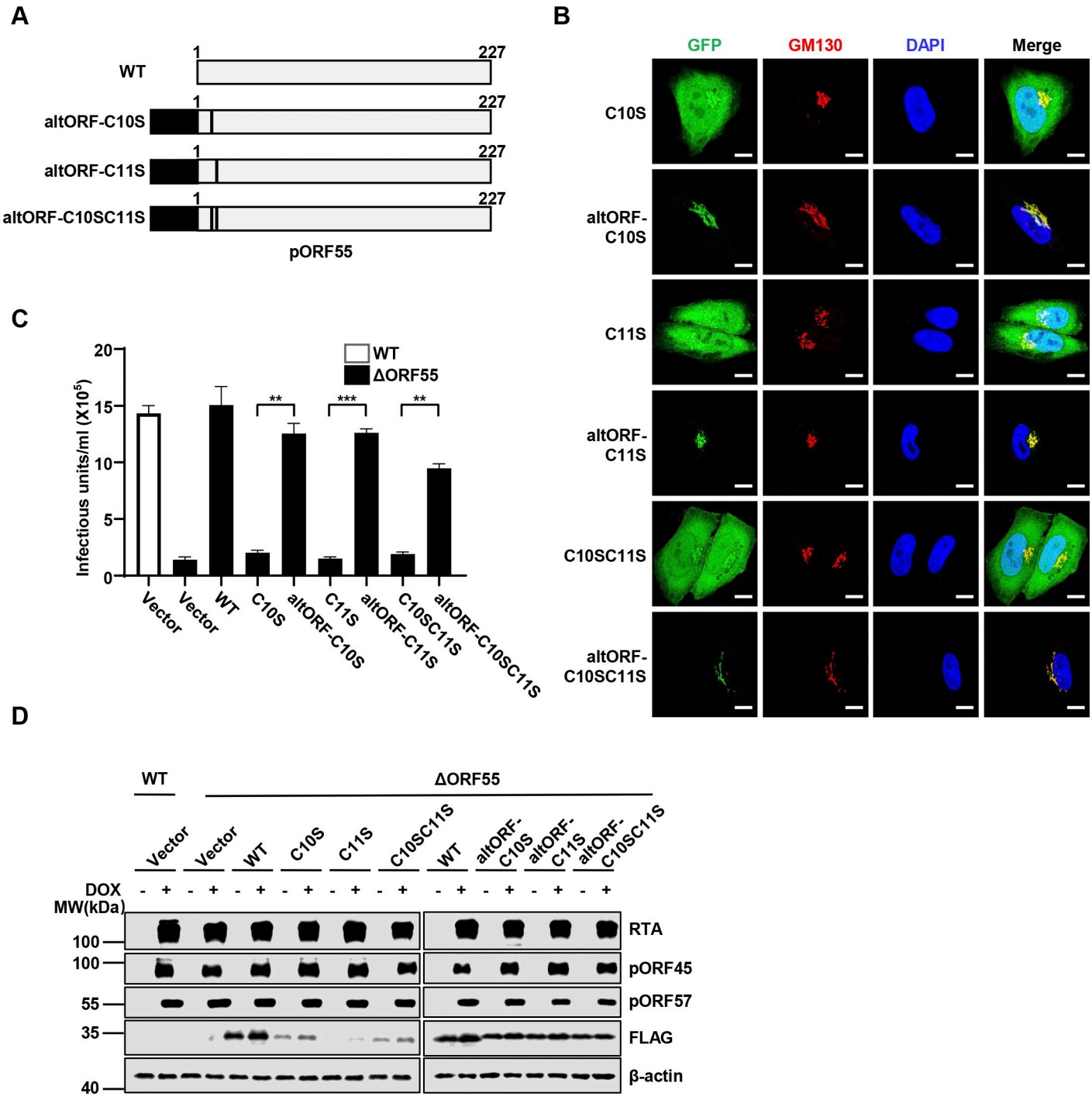

**Fig 6. The Golgi localization of pORF55 is essential for efficient virion production.** (A) Schematic diagram of pORF55 and the mutants containing altORF (a Golgi targeting sequence). (B) Hela cells expressing the indicated pORF55 mutants fused with EGFP were fixed and immunostained with GM130. The nuclei were counterstained by DAPI. Scale bars, 10 μm. (C-D) SLK.iBAC or SLK.iBACΔORF55 cells were reconstituted with vector control, FLAG-ORF55 or the indicated mutants via lentiviral transduction. The reconstituted cells were induced with Dox (1 μg/mL) and sodium butyrate (0.5 mM) for 48 h, and KSHV infectious units in the supernatants were quantified (C). WCLs were analyzed by immunoblotting (D).

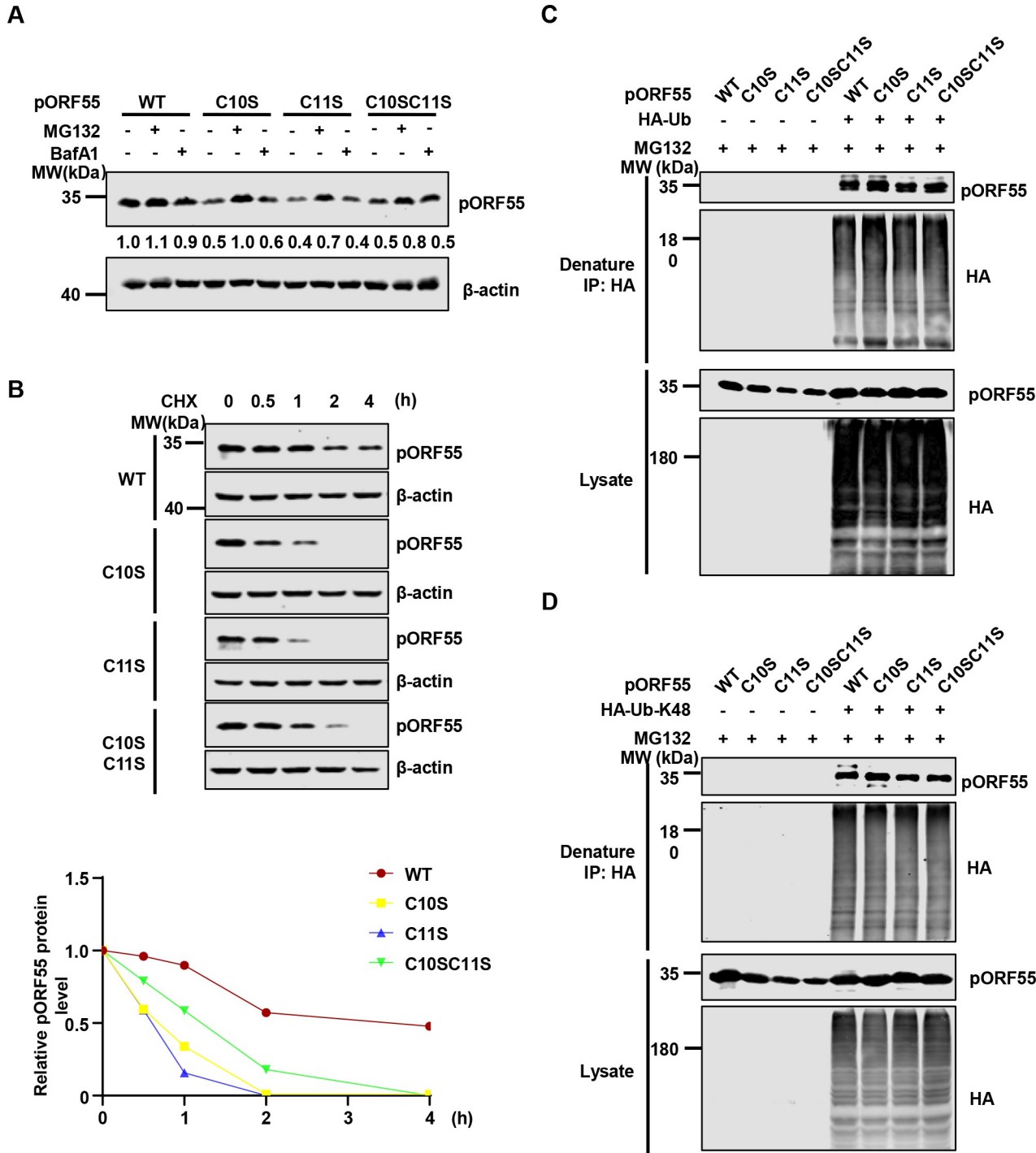

**Fig 7. The palmitoylation-defective mutans of pORF55 show reduced stability.** (A) HEK293T cells were transfected with FLAG-ORF55 or the mutants for 16 h, and were then treated with DMSO, Baf-A1 (1 mM) or MG132 (10 μM) for 10 h. WCLs were collected and analyzed by immunoblotting. (B) HEK293T cells were transfected with FLAG-ORF55 or the indicated mutants for 24 h. The cells were then treated with CHX (50 μg/mL) for the indicated time points, and WCLs were analyzed by immunoblotting. pORF55 band intensity was quantified by ImageJ, which was normalized to β-actin and to the t = 0 time point. (C) HEK293T cells were co-transfected with HA-Ub and FLAG-ORF55 or the mutants for 16 h, and then the cells were treated with MG132 (10 μM) for 10 h. Denatured immunoprecipitation was performed with anti-HA affinity agarose, followed by immunoblotting. (D) HA-Ub (K48-only), instead of HA-Ub, was used for the ubiquitination assay as described in Fig 7C.

KSHV lytic replication. Together, these data indicate that pORF55 palmitoylation promotes its stabilization and facilitates infectious virion production.

## pORF55 recruits pORF42 to the Golgi and enhances the protein level of pORF42

Previous studies reveal that pORF55 and pORF42 form a complex at the Golgi [15]. We thus sought to investigate the role of pORF55 palmitoylation in the interaction with pORF42. Co-immunoprecipitation assays indicate that pORF42 interacted with WT pORF55 and the palmitoylation-deficient mutants (S6A Fig). Immunofluorescence staining showed that pORF42-mCherry was distributed throughout the cells, with no apparent specific subcellular localization detected. However, when co-expressed with pORF55, a significant portion of pORF42 co-localized with pORF55 at the Golgi, consistent with previous reports [15]. Notably, the palmitoylation-deficient mutants of pORF55 failed to localize at the Golgi but still exhibited strong co-localization with pORF42 (S6B Fig). Interestingly, the expression of pORF55 substantially enhanced the protein level of pORF42, suggesting that pORF55 may stabilize pORF42 (S6C Fig). These data indicate that pORF55 recruits pORF42 to the Golgi and enhances the protein level of pORF42.

## Discussion

The life cycle of KSHV comprises two distinct phases, latent and lytic [7,8]. The transition from latency to lytic replication is a highly complex process that requires the coordination of a large number of viral proteins and host factors [6]. Specifically, our current understanding of the assembly process during lytic replication is very limited [30]. Recent studies have started to shed light on the critical viral proteins involved in this process. For example, ORF33, ORF38, ORF45 and ORF52 of KSHV have been identified as pivotal players in viral assembly [31–35]. Another recent study has highlighted the phase separation property of ORF52 in the formation of cytoplasmic virion assembly compartments (cVAC) and the role in the assembly process [36]. Nonetheless, the roles of numerous other KSHV lytic proteins in the assembly process remain poorly understood. Among these is pORF55, a tegument protein encoded by a late gene. ORF55 is conserved across all herpesviruses, and its homologues in HSV-1 (UL51) and EBV (BSRF1) are known to be involved in viral assembly [14,15,37], suggesting that pORF55 probably plays a critical role in KSHV viral assembly.

Among the homologues of KSHV pORF55, HSV-1 UL51 has been most extensively studied. A series of studies have revealed that UL51 is palmitoylated, and the palmitoylation of UL51 at the N-terminal Cysteine 9 is necessary for its Golgi localization [16]. UL51 is actively involved in the assembly and maturation of HSV-1 virion particles [14]. Subsequent studies have uncovered that UL51 forms a complex with UL7, and both proteins play crucial roles in viral assembly [22,37]. Structural analyses have provided valuable insights into the direct interaction between UL51 and UL7, as well as their counterparts in EBV, BSRF1 and BBRF2 [15,17]. Structural studies about BSRF1 and BBRF2 have proposed a model whereby BSRF-1 interacts with viral membrane proteins, such as gB and gH-gL, while BBRF2 associates with viral capsid protein (e.g., MCP) and major tegument proteins (e.g., BPLF1) to facilitate secondary envelopment [17]. KSHV pORF42 (a homologue of HSV-1 UL7 and EBV BBRF2) has been reported to form a complex with pORF55 at the Golgi apparatus, and is similarly required for efficient progeny virion production [15,38]. Our study here reveals that KSHV pORF55 is highly ubiquitinated, and that palmitoylation of pORF55 is necessary for its Golgi localization, which in turn stabilizes the protein, enabling it to fulfill its function in promoting secondary envelopment, viral assembly and maturation. Since we are unable to decouple the Golgi localization of pORF55 from its stabilization to investigate the function of cytoplasmically stabilized

pORF55, we conclude that the Golgi localization of pORF55 promotes its stabilization and primarily facilitates virion assembly.

A discrepancy in pORF55 stability can be observed between transfection and infection contexts (Fig 5C vs Fig 7A). During transient transfection (Fig 7A), the expression of WT pORF55 and the mutants is under the control of strong promoters (the EF1α promoter), which potentially minimizes the differences in protein levels. Nonetheless, we still found that the mutants were significantly reduced compared with WT pORF55 (Fig 7A). In contrast, during viral reactivation (Fig 5C), pORF55 and the mutants are driven by the endogenous promoter, and this experimental setting more faithfully reflects the real infection scenario. Our research has also uncovered that pORF55 plays a critical role in sustaining the protein level of pORF42, and palmitoylation of pORF55 is required to recruit pORF42 to the Golgi, where they collaboratively facilitate secondary envelopment. However, a detailed understanding of the precise roles of pORF55 and pORF42 in viral assembly warrants further investigation.

Finally, previous studies have revealed that, in addition to latency, the lytic phase also significantly contributes to KSHV-associated malignancies [39]. Notably, KSHV canonical latent infection alone cannot transform cells. It is believed that a small subset of infected cells initiates lytic reactivation, which provides essential paracrine signaling for tumorigenesis, such as promoting inflammation and angiogenesis. Moreover, the released KSHV progeny virions from the lytically infected cells can infect the neighboring cells, thereby replenishing the latently infected cell pool [39] This notion is further supported by the observations that lytic replicating cells are present in KS tumors, and that inhibition of KSHV lytic replication by antiviral treatment suppresses KSHV-associated malignancies [40]. Thus, we propose that targeting pORF55 palmitoylation to inhibit KSHV lytic replication could represent a promising therapeutic strategy for the treatment of KSHV-related diseases. First, 2-BP, a well-established palmitoylation inhibitor [41,42], efficiently blocks the palmitoylation of pORF55 and disrupts its Golgi localization, and 2-BP treatment strongly inhibits KSHV replication. These findings suggest that it is possible to specifically target pORF55 palmitoylation to curtail viral replication. Second, the successful design of palmitoylation inhibitors targeting specific protein palmitoylation supports this approach. For example, Stimulator of interferon genes (STING) is essential for the induction of antiviral innate immune responses against intracellular DNA [43,44]. STING is palmitoylated at Cysteine 88/91 at the Golgi, which is necessary for its activation [45]. Subsequently, a small molecular inhibitor, H-151, was developed to covalently modify STING at cysteine 91, effectively blocking STING signaling [46]. H-151 has demonstrated efficacy in various STING-related disease models [47–49]. Therefore, we propose that targeting viral protein palmitoylation, rather than promiscuous palmitoylation inhibitors (e.g., 2-BP), has the potential to reduce non-specific cell toxicity and selectively hinder viral replication.

In summary, KSHV pORF55 is palmitoylated at Cysteine 10 and 11, and the palmitoylation of pORF55 is required for its Golgi localization and secondary envelope formation. Palmitoylation-deficient mutants of pORF55 are unstable and undergo proteasomal degradation. Forced redirecting the palmitoylation-defective mutants of pORF55 to the Golgi leads to their stabilization and is able to restore impaired progeny virion production. Together, our study provides new insight into the critical role of pORF55 palmitoylation in KSHV infection and imply that targeting viral palmitoylation could serve as a potential antiviral approach.

## Materials and methods

### Cell culture

SLK [50] (kindly provided by Dr. Pinghui Feng, University of Southern California), HEK293T, and Hela cells were cultured in DMEM (Sigma) supplemented with 10% fetal

bovine serum (FCS) (Lonsera, Shuangru Biotech, Shanghai, China) and 1% penicillin-streptomycin (HyClone). SLK.iBAC cells [21,35] were kindly provided by Dr. Fanxiu Zhu (Florida State University) and were cultured in DMEM supplemented with 10% FCS, 1% penicillin-streptomycin and hygromycin (500 μg/mL) (Invivogen).

## Antibodies and reagents

Antibodies for immunoblotting: Rabbit polyclonal anti-KSHV pORF55 (1:1000) and Rabbit polyclonal anti-KSHV RTA (1:1000) were generated by Dia-An Biotechnology (Wuhan, China); Mouse monoclonal anti-KSHV pORF45 (sc-53883; 1:500) and Mouse monoclonal anti-KSHV pORF57 (sc-135746; 1:1000) were from Santa Cruz; Mouse monoclonal anti-β-actin (catalog no. 2060; 1:2000) and Mouse monoclonal anti-FLAG (catalog no. 2064; 1:5000) were from Dia-An Biotechnology (Wuhan, China); Mouse anti-HA monoclonal antibody (Dia-An Biotechnology; catalog no. 2063; 1:3000); IRDye 800CW goat anti-rabbit (1:1000) and goat anti-mouse (1:1000) secondary antibodies were from Li-Cor.

Antibodies for immunofluorescence: Rabbit polyclonal anti-KSHV pORF55 (1:200); Mouse anti-GM130 (AB-398142; BD Biosciences; 1:200); Mouse monoclonal anti-calnexin (sc-23954; Santa Cruz; 1:100); Rabbit monoclonal anti-GM130 (A11408; ABclonal; 1:200); Alexa Fluor 594 goat anti-mouse (A-11032; 1:1000) and anti-rabbit (A-11012; 1:1000) were from Invitrogen; Alexa Fluor 488 goat anti-rabbit (A-11008; 1:1000; Invitrogen).

2-Bromohexadecanoic acid (2-BP), Azo biotin-azide, Doxycycline (Dox) and sodium butyrate were purchased from Sigma; Alkynyl-palmitic-acid (Alk-C16) and THPTA were purchased from Click Chemistry Tools; Puromycin, hygromycin and blasticidin were purchased from invivogen; Streptavidin agarose resin was purchased from Invitrogen; Cycloheximide (CHX), Bortezomib and Bafilomycin-A1 (Baf-A1) were purchased from Selleck; MG132 and Ganciclovir were ordered from MedChemExpress; LDH Cytotoxicity Assay Kit was purchased from Beyotime (Shanghai, China).

## Constructs

sgRNAs targeting the indicated viral genes were constructed into Lenti-CRISPRv2, and the gRNA sequences are listed in S1 Table. ORF55 was amplified from KSHV BACmid [20,21] and subcloned into pEF-FLAG-N, pEF-EGFP and LentiBlast-FLAG vectors. The palmitoylation-defective mutants of ORF55 were generated via site-directed mutagenesis. altORF was inserted into the N-terminus of ORF55-C10S, -C11S and -C10SC11S by homologous recombination.

## Generation of KSHV mutants and quantification of infectious units

KSHVΔORF55, KSHV-ORF55-C10S, KSHV-ORF55-C11S, KSHV-ORF55-C10SC11S were generated by two-step Red-mediated recombination as previously described [19,51]. The primer sequences are listed in S2 Table. KSHV BACmids (iBAC-WT, iBACΔORF55, iBAC-ORF55-C10S, iBAC-ORF55-C11S and iBAC-ORF55-C10SC11S) were extracted from E. coli GS1783 and transfected into SLK cells with Fugene HD (Promega, Wisconsin, United States) to generate stable cells.

SLK.iBAC cells were treated with Dox (1 μg/mL) and sodium butyrate (0.5 mM) for 48 h to induce lytic reactivation, and the supernatants were collected to infect HEK293T cells. Then the cells were trypsinized at 24 h post-infection, fixed with 1% (W/V) paraformaldehyde (PFA) (#DF0131, LEAGENE, Beijing, China), and subjected to flow cytometry analysis. KSHV infectious units were quantified based on GFP-positive cell percentage, and the flow cytometry data were analyzed with FlowJo 10.0 [52].

## Stable cell line generation

Lentiviruses containing the indicated genes or sgRNAs were generated in HEK293T cells as previously described [53,54]. SLK.iBAC cells were spin-infected (500 g, 50 min) with lentiviruses in the presence of polybrene (8 μg/mL). After 48 h post-infection, the transduced cells were selected with appropriate antibiotics for 3 days. SLK.iBAC cells containing sgRNA targeting the indicated viral genes were selected with puromycin (1 μg/mL) and hygromycin (500 μg/mL), while SLK.iBAC cells stably expressing FLAG-pORF55 and the mutants were selected with blasticidin (10 μg/mL) and hygromycin (500 μg/mL).

## RNA extraction and RT-qPCR

SLK.iBAC cells were induced with Dox (1 μg/mL) for 24 h. Total RNA was extracted using TRIzol reagent (Takara), and cDNA was synthesized using HiScriptII 1st strand cDNA synthesis Kit (Vazyme, Nanjing, China). The cDNA mixture was diluted 20-times and subjected to qPCR analysis with SYBR green qPCR master mix (Bimake, Shanghai, China). The relative gene expression was normalized to ATCB. The primer sequences are listed in S2 Table.

## Quantification of viral genome copy number

SLK.iBAC cells were treated with Dox (1 μg/mL) and sodium butyrate (0.5 mM) for 48 h to induce lytic reactivation. Genomic DNA in the supernatants was extracted by phenol-chloroform extraction as previously described [52]. Intracellular DNA was extracted with TIANamp Genomic DNA Kit (DP304, TIANGEN, Beijing). The viral genomic DNA was quantified by qPCR, and a stand curve was generated using serial dilutions of pEF-FLAG-RTA plasmid [52].

## Immunofluorescence

Hela cells were transfected the indicated plasmids, and SLK.iBAC cells were induced with Dox (1 μg/mL) for 24h. Then the cells were washed three times with PBS and fixed with 4% PFA for 20 min. Subsequently, the cells were permeabilized with 1% Triton X-100 for 10 min and blocked with 10% goat serum (Antgene, Wuhan, China) for 1 h at room temperature. Next, Hela cells were incubated with rabbit anti-GM130 polyclonal antibody (1:200; A11408; ABclonal) or mouse anti-calnexin monoclonal antibody (1:100; sc-23954; Santa Cruz), while SLK.iBAC cells were incubated with rabbit anti-pORF55 polyclonal antibody (1:200) and mouse anti-GM130 monoclonal antibody (1:200; AB-398142; BD Bioscience) overnight at 4°C.Then the cells were washed with PBST (Phosphate Buffered Saline with 0.05% Tween-20, pH 7.4) for three times, and were incubated with the secondary antibodies for 90 min at room temperature. After washing with PBST, the slides were mounted with DAPI Fluoromount-G mounting medium (SouthernBiotech). The images were collected with a confocal microscope (Zeiss LSM880), and the pictures were processed using ZEN 2 (Zeiss) and ImageJ (NIH).

## Metabolic labeling and click chemistry to detect protein palmitoylation

For metabolic labeling, HEK293T cells were transfected with FLAG-ORF55 or the mutants, and SLK.iBAC cells were induced with Dox (1 μg/mL) for 16 h. Then DMSO or Alkynyl-palmitic-acid (Alk-C16; 100 μM) was added to the culture medium and incubated for an additional 8 hours.

The labeled cells were lysed with 800 μL of lysis buffer (1% Triton-X-100, 0.1% SDS, 50 mM Tris-HCl pH 7.4, 150 mM NaCl) supplemented with protease inhibitors for 20 min at 4°C. Cell lysates were centrifuged at 12,000 rpm for 15 min at 4°C, and the supernatants were collected. Azo biotin-azide (100 μM), THPTA (1 mM), CuSO4 (1 mM), and sodium ascorbate

(10 mM) were added to the supernatants to initiate the CuAAC reaction at room temperature. After 1 h, the reaction was terminated by adding a nine-fold volume of ice-cold methanol, and the reaction tubes were incubated at -80˚C overnight. The following day, the protein pellet was collected by centrifugation at 12,000 rpm for 10 min at 4˚C, and washed twice with ice-cold methanol, then air-dried. Proteins were dissolved by adding 200 μL of dissolution buffer (50 mM Tris-HCl pH 7.4, 150 mM NaCl, 5 mM EDTA, 2% SDS).

Subsequently, the dissolved proteins were diluted 20-fold with wash buffer (50 mM Tris-HCl pH 7.4, 150 mM NaCl, 5 mM EDTA, 0.5% NP40) to reduce the concentration of SDS to 0.1%, and immunoprecipitation was performed to isolate labeled proteins [55]. Prewashed streptavidin agarose resins were then added to the solution, and the mixture were incubated for 4 h at 4˚C. The streptavidin resins were collected by centrifugation and extensively washed with wash buffer. The precipitated proteins were released by boiling in SDS sample buffer for 15 min at 95˚C, separated by SDS-PAGE, and detected by immunoblotting.

## Cycloheximide (CHX) Chase Assay

HEK293T cells were transfected with FLAG-ORF55 or the mutants. Sixteen hours post-transfection, the cells were treated with cycloheximide (CHX; 50 μg/mL). Cells were harvested at the indicated time points, and lysed for immunoblotting analysis. The protein band densities were quantified by ImageJ (NIH) and normalized to β-actin.

## Immunoprecipitation

HEK293T cells were transfected with HA-ORF42 and FLAG-ORF55-GFP and cells were collected and lysed in lysis buffer (150 mM NaCl, 50 mM Tris-HCl pH 7.4, 1% Triton X-100, 1 mM EDTA) supplemented with protease inhibitors at 24 h post-transfection. The supernatants were collected and incubated with anti-FLAG beads (Dia-An Biotechnology, Wuhan, China) at 4˚C for 4 h. The resins were collected by centrifugation and extensively washed with wash buffer. The precipitated proteins were released by boiling in SDS sample buffer for 15 min at 95˚C, separated by SDS-PAGE, and detected by immunoblotting.

## Ubiquitination assay

HEK293T cells were co-transfected with FLAG-ORF55 or the mutants and HA-ubiquitin. Sixteen hours post-transfection, cells were treated with MG132 (10 μM) for 10 h. The cells were then lysed with lysis buffer (150 mM NaCl, 50 mM Tris-HCl pH 7.4, 1% Triton X-100, 1 mM EDTA) supplemented with protease inhibitors. The collected supernatant was supplemented with 1% SDS and boiled for 10 min at 95˚C. Next, the solution was diluted 10-fold to reduce the concentration of SDS to 0.1%, followed by immunoprecipitation with anti-HA agarose as described above.

## Transmission Electron Microscopy (TEM)

SLK.iBAC cells were treated with Dox (1 μg/mL) and sodium butyrate (0.5 mM) for 3 days to induce lytic reactivation. Then the cells were fixed with 2.5% glutaraldehyde at room temperature for 2 hours. Subsequently, the cells were gently scraped and collected by centrifugation. The collected cells were washed three times with 0.1 M PBS. Following this, the cells were fixed with 1% osmium tetroxide pre-cooled to 4˚C for 2–3 hours at 4˚C and underwent another three washes with 0.1 M PBS. The samples were dehydrated through a series of gradient ethanol concentrations (50%, 70%, 80%, 85%, 90%, 95%, 100%), with each concentration applied for 15 minutes. Then the samples were infiltrated with a mixture of acetone and epoxy resin

(2:1), followed by acetone and epoxy resin (1:1), and epoxy resin for 12 hours at 37˚C. Lastly, the samples were embedded and sectioned, and the images were collected with a Transmission Electron Microscope (Tecnai G2 T20, 200 kV).

## Whole-genome sequencing of KSHV BACs

KSHV BACmids (iBAC-WT, iBACΔORF55, iBAC-pORF55-C10S, iBAC-pORF55-C11S and iBAC-pORF55-C10SC11S) were extracted from E. *coli* GS1783. The sequencing was conducted by ANOROAD gene technology company (Beijing, China). Trimmomatic was utilized to filter out low-quality reads [56]. The high-quality sequence reads were aligned to the reference genome (GenBank: GCA_027939395.1) using BWA-mem with default settings [57]. SAMtools was employed for viewing, sorting, and indexing the alignment results [58]. Subsequently, Gatk was utilized for variant calling [59]. Ultra-high coverage (>1500×) was achieved for each mutation, enabling comprehensive whole-genome sequence analysis (S3 Table). IGV (Integrative Genomics Viewer) was employed for visualization [60].

## Statistical analysis

GraphPad Prism (version 8) was used to analyze all statistical data. Data represent the mean of at least three independent experiments, and error bars denote standard deviation (S.D.). Two-tailed student's t test or analysis of variance (ANOVA) was used for statistical analysis. Significant differences are represented by p value (*$p < 0.05$, **$p < 0.01$, ***$p < 0.001$).

## Supporting information

**S1 Fig. Palmitoylation of pORF55 is necessary for its Golgi localization.** (A) SLK.iBAC or SLK.iBACΔORF55 were induced with Dox (1 μg/mL) and sodium butyrate (0.5 mM) to trigger lytic reactivation. SLK.iBAC cells were treated with ganciclovir or 2-BP (10 μM, 30 μM, 50 μM, and 80 μM). KSHV infectious units were quantified at 48 h post-induction. (B) SLK. iBAC cells were incubated with DMSO or the indicated inhibitors for 48 h, and cell viability was quantified by the LDH release assay. (C) Schematic diagram of the pORF55 mutants. (D) Hela cells were transfected with the indicated plasmids, and immunofluorescence staining was performed with an antibody against GM130. The nuclei were counterstained by DAPI. Scale bars,10 μm. (E) Hela cells were transfected with ORF55-EGFP or the mutants, and immunofluorescence staining were performed with an antibody against GM130 (a Golgi marker). The nuclei were counterstained by DAPI. Scale bars, 50 μm.
(TIF)

**S2 Fig. pORF55, but not the palmitoylation-defective mutants, rescues KSHVΔORF55 virion production.** (A-E) SLK.iBAC or SLK.iBACΔORF55 cells were reconstituted with vector control, FLAG-pORF55 or the indicated mutants via lentiviral transduction. The reconstituted cells were induced with Dox (1 μg/mL) and sodium butyrate (0.5 mM) for 48 h, and KSHV infectious units in the supernatants were quantified (A). Viral gene transcription was quantified by RT-qPCR (B), and WCLs were analyzed by immunoblotting (C). The intracellular and extracellular viral genome copy number was determined by qPCR analysis (D and E)
(TIF)

**S3 Fig. Palmitoylation of pORF55 is required for secondary envelopment.** SLK.iBAC cells containing WT ORF55, ΔORF55 or the indicated mutants were induced with Dox (1 μg/mL) and sodium butyrate (0.5 mM) for 3 days, and the images were acquired by Transmission Electron Microscopy. Scale bars, 500 nm.
(TIF)

**S4 Fig. altORF restores the Golgi localization of the pORF55 mutant.** (A) Hela cells were transfected with ORF55-EGFP or altORF-EGFP, followed by immunofluorescence staining with GM130 as a Golgi marker. (B) Hela cells were transfected with ORF55ΔN15-EGFP or altORF-ORF55ΔN15-EGFP, followed by immunofluorescence staining with GM130 as a Golgi marker. (C) Hela cells expressing the indicated ORF55 mutants fused with EGFP were fixed and immunostained with GM130. The nuclei were counterstained with DAPI. Scale bars,10 μm.
(TIF)

**S5 Fig. Treatment with proteasome inhibitors suppressed KSHV lytic replication.** (A) SLK. iBAC cells containing WT ORF55, ΔORF55 or the indicated mutants were treated with DMSO, MG132 (10 μM), or Bortezomib (100nM) for 10 h, followed by induction with Dox (1 μg/mL) and sodium butyrate (0.5 mM) to trigger lytic reactivation. KSHV infectious units were quantified at 48 h post-induction. (B) SLK.iBAC cells were treated as described in S5A Fig. WCLs were collected at 24 h post-induction and analyzed by immunoblotting.
(TIF)

**S6 Fig. pORF55 recruits pORF42 to the Golgi and enhances the protein level of pORF42.** (A) HEK293T cells were co-transfected with HA-ORF42 and FLAG-ORF55-GFP or the mutants, and WCLs were collected for immunoprecipitation with anti-FLAG affinity agarose. The input and precipitated samples were analyzed by immunoblotting. (B) Hela cells were co-transfected with ORF42-mCherry and ORF55-GFP or the mutants, and immunofluorescence staining were performed with an antibody against GM130 (a Golgi marker). The nuclei were counterstained by DAPI. Scale bars,10 μm. (C) HEK293T cells were transfected with HA-ORF42 and FLAG-ORF55 as indicated, followed by immunoblotting analysis.
(TIF)

**S1 Table. sgRNA information.**
(XLSX)

**S2 Table. Primer information.**
(XLSX)

**S3 Table. Whole-genome sequencing analysis.**
(XLSX)

**S4 Table. Raw data.**
(XLSX)

## Acknowledgments

We thank Drs. Pinghui Feng (University of Southern California), Fanxiu Zhu (Florida State University), Jae Jung (Cleveland Clinic) and Kevin Brulois (Stanford University) for reagents, and Drs. Lang Bu and Jianping Guo (Sun Yat-sen University) for suggestions. We thank Pei Zhang and An-Na Du (the core facility and technical support, Wuhan Institute of Virology) for their help with TEM. We thank the core facility of the Medical Research Institute at Wuhan University for excellent technical support.

## Author Contributions

**Conceptualization:** Ke Lan, Junjie Zhang.

**Formal analysis:** Yaru Zhou, Shaowei Wang, Jiali Ma, Lei Bai, Min-Hua Luo, Qingsong Qin, Baishan Jiang, Ke Lan, Junjie Zhang.

**Funding acquisition:** Qingsong Qin, Junjie Zhang.

**Investigation:** Yaru Zhou, Xuezhang Tian, Shaowei Wang, Ming Gao, Chuchu Zhang, Xi Cheng.

**Methodology:** Hai-Bin Qin, Min-Hua Luo.

**Resources:** Lei Bai, Baishan Jiang.

**Supervision:** Ke Lan, Junjie Zhang.

**Writing – original draft:** Yaru Zhou, Junjie Zhang.

**Writing – review & editing:** Yaru Zhou, Junjie Zhang.

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
