## [Decision Letter · Decision Letter 0]

18 Dec 2023

Dear Dr. Zhang,

Thank you very much for submitting your manuscript "Palmitoylation of KSHV ORF55 is required for Golgi localization and efficient progeny virion production" for consideration at PLOS Pathogens. As with all papers reviewed by the journal, your manuscript was reviewed by members of the editorial board and by several independent reviewers. In light of the reviews (below this email), we would like to invite the resubmission of a significantly-revised version that takes into account the reviewers' comments.

We cannot make any decision about publication until we have seen the revised manuscript and your response to the reviewers' comments. Your revised manuscript is also likely to be sent to reviewers for further evaluation.

Sincerely,

Shou-Jiang Gao, Ph.D.

Academic Editor

PLOS Pathogens

Patrick Hearing

Section Editor

PLOS Pathogens

Kasturi Haldar

Editor-in-Chief

PLOS Pathogens

orcid.org/0000-0001-5065-158X

Michael Malim

Editor-in-Chief

PLOS Pathogens

orcid.org/0000-0002-7699-2064

Reviewer's Responses to Questions

**Part I - Summary**

Reviewer #1: The manuscript by Zhou et al. describes the authors investigations into the role of the conserved tegument protein encoded by ORF55 of Kaposi’s sarcoma-associated herpesvirus. The rationale of this study is that many of the structural proteins of the virus have not been fully characterized as far as their function and activities in the infected cell. These lytic gene products the authors propose can lead to cellular transformation by the virus following reactivation from the latent state. Previous studies by the Jung Lab have shown that the vBcl2 can peturb the incorporation of ORF55 into the virion particle and this identified ORF55 as being essential for replication. ORF55 is conserved in all herpesviruses indicating the importance of this protein. The homologue in HSV-1 has been studied the most. That gene encoded by UL51 is an important structural protein and displays alternate replication strategies in cells when partnered with UL7. This complex is required for secondary envelopment, egress and cell to cell spread. The HSV-1 protein has been shown to be palmitoylated and this modification is required for it to localize to the Golgi.

In this study the authors demonstrate ORF55 is required for virus production by making a BAC knockout in the gene. They also show ORF55 is palmitoylated and this plays a role in Golgi localization of the protein. Mutation in the ORF55 cysteines that are palmitoylated abolish this modification and consequently Golgi localization. The authors use a cool trick of adding a Golgi localizing peptide to these mutants and thereby restoring Golgi localization and function. They also show that mutant ORF55 that are not Golgi localized are more unstable. The data presented are of high-quality including controls and thorough examination of virus mutants using infected and transfected cells. My main concern is what is the novelty here that we do not already know especially for this journal.

Reviewer #2: This manuscript by Zhou et al describes the study on ORF55 encoded by KSHV. The HSV-1 homologue of ORF55, UL51, has been extensively investigated. While the presented results are clear and convincing, they largely agree with the known information about UL51, for instance, the importance of palmitoylation for Golgi localization and its role in virion production. The results described in this study are not surprising given that ORF55 is one of the highly conserved structural proteins involved in virion assembly. While the conservation of structural proteins is noteworthy, the manuscript falls short in providing significant advancements or new insights beyond existing knowledge. The study also highlights the antiviral effects of palmitoylation inhibitor, BP-2, but overall, the contribution of the manuscript regarding novelty and uniqueness appears limited.

Reviewer #3: In this interesting manuscript, Zhou and colleagues show that the KSHV tegument protein pORF55 is required for virus assembly and/or release in the late stages of the productive viral replication cycle and that its recruitment to Golgi membranes is essential for this role and depends on palmitoylation of two cysteine residues in the pORF55 N-terminal domain. This observation mirrors a similar mechanism previously reported for the equivalent HSV tegument protein. Technically, the reported experimental evidence is clear-cut; in the experiment shown in figure 7 C,D it might have been better to have immunoprecipitated the tagged ubiquitin and then blotted for pORF55 mutants in order to obtain clear-cut bands of ubiquitinated proteins.

**Part II – Major Issues: Key Experiments Required for Acceptance**

Reviewer #1: The study could be strengthened by incorporating experiments that examine the interaction with ORF42. This complex is important for the activities and functions of both proteins. See lines 238 to 241.

What is the fate of the virus in the ORF55 KO cell lines? This would be important to visualize using electron microscopy or imaging viral capsids in the confocal.

Were all the KSHV mutants sequenced by whole genome sequencing?

Reviewer #2: 1. It will be informative to include an inhibitor of DNA replication as a reference for the extent of the impact on virion production from the absence of ORF55 and BP-2.

2. The statement, "However, the inhibitory effect was greatly diminished when 2-BP was applied to SLK.iBACΔORF55 cells (Figure 4B)." needs careful consideration. Drawing conclusions from the negative data in Figure 4 is challenging due to the already low virion production of the ORF55 mutant. It is plausible that the absence of ORF55 might obscure other palmitoylation events critical for virion production.

3. There appears to be a difference in ORF55 stability between transfection and in the context of infection. While the difference between wild-type and mutant ORF55 is minimal in the transfection setting (Fig. 7A), it becomes more pronounced in the context of infection (Fig. 5C)." Please clarify.

Reviewer #3: 1. Figure 7C,D: the bands representing ubiquitinated proteins appear very 'smeary' on this blot. It may be better to immunoprecipitate with an antibody to the HA tag on the transfected ubiquitin and then blot for the pORF55 mutants - this may show the ubiquitinated pORF55 proteins more clearly.

**Part III – Minor Issues: Editorial and Data Presentation Modifications**

Reviewer #1: Many of the immunofluorescence images only show one cell. Were these representative of the whole culture?

Reviewer #2: (No Response)

Reviewer #3: 1. lines 126/127 and 245/246: the authors state that "the inhibitory effect of 2-BP was considerably diminished" in the case of the KSHV ORF55 deletion mutant. Since this mutant hardly produces any infectious progeny anymore (Figure 4), there can be no further inhibition of viral progeny formation by the 2-BP compound and it may therefore be better to rephrase these two sentences.

2. The authors should use the term "ORF55" when they refer to the gene, and "pORF55" or "ORF55 protein" when they refer to the protein.

PLOS authors have the option to publish the peer review history of their article (what does this mean?). If published, this will include your full peer review and any attached files.

Reviewer #1: No

Reviewer #2: No

Reviewer #3: **Yes: **Thomas F. Schulz, MD
---

## [Decision Letter · Decision Letter 1]

29 Feb 2024

Dear Dr. Zhang,

Thank you very much for submitting your manuscript "Palmitoylation of KSHV pORF55 is required for Golgi localization and efficient progeny virion production" for consideration at PLOS Pathogens. As with all papers reviewed by the journal, your manuscript was reviewed by members of the editorial board and by several independent reviewers. The reviewers appreciated the attention to an important topic. Based on the reviews, we are likely to accept this manuscript for publication, providing that you modify the manuscript according to the review recommendations.

Sincerely,

Shou-Jiang Gao, Ph.D.

Academic Editor

PLOS Pathogens

Patrick Hearing

Section Editor

PLOS Pathogens

Michael Malim

Editor-in-Chief

PLOS Pathogens

orcid.org/0000-0002-7699-2064

Reviewer Comments (if any, and for reference):

Reviewer's Responses to Questions

**Part I - Summary**

Reviewer #1: I have reviewed all the suggestions and experiments required and I am happy to say that they have been addressed.

I will thus suggest to accept the revised manuscript.

Reviewer #2: The observation that Golgi localization stabilizes ORF55 is intriguing; however, this localization primarily facilitates its role in virion assembly. Even if a proteasome inhibitor stabilizes the palmitoylation mutant, it may not rescue virion production. While the stability observation is novel, it does not significantly advance our understanding of ORF55 and its homologs beyond its function in virion assembly.

Moreover, targeting palmitoylation presents a potential strategy for HSV-1 diseases resulting from lytic replication. However, its broad impact on cellular proteins raises concerns about toxicity, unlike the specificity offered by acyclovir. The relevance of blocking KSHV virion production in treating diseases primarily associated with viral latency remains unclear.

Reviewer #3: The authors have addressed the reviewers' comments in a satisfactory manner. The manuscript essentially shows that KSHV pORF55 plays a similar role during secondary envelopment and virus production as its homologue in HSV1. Technically, the presented experiments are of high quality. The authors have added additional results, in particular relating to the recruitment of pORF42 by pORF55 and the relationship between palmytoylation and ubiquitination of pORF55.

**Part II – Major Issues: Key Experiments Required for Acceptance**

Reviewer #1: (No Response)

Reviewer #2: (No Response)

Reviewer #3: none

**Part III – Minor Issues: Editorial and Data Presentation Modifications**

Reviewer #1: (No Response)

Reviewer #2: (No Response)

Reviewer #3: none

PLOS authors have the option to publish the peer review history of their article (what does this mean?). If published, this will include your full peer review and any attached files.

Reviewer #1: No

Reviewer #2: No

Reviewer #3: No

Figure Files:

Data Requirements:

Reproducibility:

References:

---

## [Editor Report · Decision Letter 2]

22 Mar 2024

Dear Dr. Zhang,

We are pleased to inform you that your manuscript 'Palmitoylation of KSHV pORF55 is required for Golgi localization and efficient progeny virion production' has been provisionally accepted for publication in PLOS Pathogens.

Best regards,

Shou-Jiang Gao, Ph.D.

Academic Editor

PLOS Pathogens

Patrick Hearing

Section Editor

PLOS Pathogens

Michael Malim

Editor-in-Chief

PLOS Pathogens

orcid.org/0000-0002-7699-2064
---

## [Editor Report · Acceptance letter]

7 Apr 2024

Dear Dr. Zhang,

We are delighted to inform you that your manuscript, "Palmitoylation of KSHV pORF55 is required for Golgi localization and efficient progeny virion production," has been formally accepted for publication in PLOS Pathogens.

Best regards,

Michael Malim

Editor-in-Chief

PLOS Pathogens

orcid.org/0000-0002-7699-2064